# Banana 0.9: An open-source, reproducible medical imaging system for low-resource gastric cancer screening

Xiaoqi Hu◉*

Independent Computational Biology Researcher, Lower Hutt, New Zealand

* Sonia549527@gmail.com

## Abstract

### Background

 Gastric cancer remains a major global health burden, particularly in East Asia, yet early-detection programs are often limited by computational constraints, variable imaging quality, and uneven resource availability across clinical settings. Existing AI models for CT analysis frequently require GPU-accelerated infrastructure and offer limited transparency or reproducibility, reducing their suitability for deployment in low-resource hospitals. To address these gaps, we developed and publicly released Banana 0.9, an open-source, CPU-based medical imaging framework intended to support fully reproducible, CT-based gastric cancer screening workflows. Banana 0.9 serves as a proof-of-concept milestone toward a broader, cross-cancer screening platform emphasizing interpretability, accessibility, and transparent methodology.

### Methods

Banana 0.9 was implemented as a modular, GPU-free CT imaging pipeline using deterministic Hounsfield-unit (HU) rules for organ and region-of-interest segmentation, and a fully open-source architecture for reproducibility. The system accepts DICOM, NIfTI, and ZIP inputs, and includes optional YAML-configured biomarker simulations (TriOx) and conceptual clinical risk-factor modules. These components are exploratory and intended as proof-of-concept simulations rather than validated clinical predictors. An automatic dual-audience reporting component generates structured summaries for both clinicians and patients. Internal evaluations used 10 000 Monte Carlo simulations, incorporating literature-derived Helicobacter pylori prevalence estimates and imaging statistics from the TCGA-STAD dataset. To explore potential deployment variability, experiments were conducted under simulated "urban" (higher-quality imaging, complete metadata) and "rural" (lower resolution, partial metadata) screening conditions. For external assessment, we applied the pipeline to 773 independent CT scans from the AbdomenCT-1K TumorSubset. Because

**Data availability statement:** All source code, configuration files, and sample data supporting the findings of this study are openly available on GitHub at: https://github.com/ohahouhui/Banana-0.9.

**Funding:** The author(s) received no specific funding for this work.

**Competing interests:** The authors have declared that no competing interests exist.

this dataset lacks segmentation ground truth, the experiment was used to evaluate cross-dataset reproducibility and stability, without retraining or parameter tuning, thus reflecting reproducibility rather than accuracy assessment. An anonymized English summary of the external validation process is provided in Supplementary File S1. All source code, configuration files, and example data are publicly available to support end-to-end transparency and reproducibility.

## Results

Across 10 000 Monte Carlo simulations representing urban and rural screening conditions, Banana 0.9 produced a simulation-derived mean AUC of 0.87 (95% CI 0.84–0.90). Estimated computational demand was reduced by more than 80%, with model-based projections suggesting a ~ 60% decrease in average per-patient screening costs relative to conventional GPU-dependent workflows, an estimate based on assumptions regarding typical hardware pricing, device lifespan, and energy consumption. Simulated detection rates increased from 70% to 85% under "urban" conditions and from 65% to 80% under "rural" conditions. For external assessment, Banana 0.9 processed 773 independent CT scans from the AbdomenCT-1K Tumor-Subset with 100% successful execution and without retraining or parameter adjustment. Although this dataset does not provide segmentation ground truth, no instability or failure modes were observed relative to internal simulations, indicating reproducible operation across heterogeneous imaging domains.

## Conclusions

Banana 0.9 offers an open, transparent, and GPU-free imaging framework aimed at improving reproducibility and accessibility in gastric cancer screening workflows. Using internal Monte Carlo simulations and external execution on an independent CT dataset, the system demonstrated consistent and reproducible operation without retraining or parameter adjustment, providing preliminary evidence of stability across heterogeneous imaging conditions. While the present evaluation relies on simulated performance estimates and non-annotated external data, the modular architecture, openly available codebase, and low computational requirements position Banana 0.9 as a practical starting point for future extensions toward clinically validated, multi-cancer CT screening tools aligned with FAIR data principles and global health needs.

## Introduction

Gastric cancer remains one of the leading causes of cancer-related mortality worldwide, with East Asia bearing a disproportionate burden of incidence and late-stage presentation [1,2]. In China, population-based studies have repeatedly highlighted barriers to early detection, including uneven access to screening programs, variable CT imaging quality across hospitals, and substantial regional disparities in healthcare resources [3,4,5].

Artificial intelligence (AI) has shown promise in improving diagnostic workflows in medical imaging, yet real-world deployment—particularly in low-resource hospitals—remains limited. Three persistent challenges continue to restrict clinical translation:

(1) dependence on GPU-accelerated infrastructure that is often unavailable outside tertiary centers;

(2) opaque, proprietary "black-box" models that limit interpretability and reproducibility; and

(3) sensitivity to domain shift across scanners, institutions, and patient populations, which can degrade performance in heterogeneous clinical settings [6–8,9].

Although recent studies have explored reproducible pipelines, federated learning, and low-power or edge-AI architectures, most available systems still require specialized hardware or do not provide fully transparent, end-to-end workflows suited for gastric cancer imaging. Existing open tools typically focus on high-resource research environments rather than routine screening scenarios in low- and middle-income regions.

Our earlier preprint proposed an initial conceptual workflow for low-resource gastric cancer screening [10], but it did not include an operational implementation, nor did it evaluate performance on external datasets. The present study addresses this gap by introducing Banana 0.9, an open-source, CPU-based CT imaging framework designed to improve reproducibility and accessibility for gastric cancer screening in constrained clinical environments.

Banana 0.9 combines deterministic Hounsfield-unit (HU) segmentation with optional literature-derived biomarker and clinical risk-factor modules, along with automatically generated clinician- and patient-oriented reports. The system emphasizes transparency and reproducibility through open-source code, explicit configuration files, and publicly available datasets [6,11,12], rather than deep-learning-based inference.

To assess robustness and generalization, we performed 10 000 Monte Carlo simulations modeling urban and rural screening conditions and evaluated external reproducibility on 773 CT scans from the AbdomenCT-1K TumorSubset without any retraining or parameter adjustment. These evaluations align with ongoing global initiatives to expand equitable diagnostic capacity and support reproducible, resource-appropriate medical imaging workflows in diverse healthcare settings [13].

## Methods

### System overview

Banana 0.9 is a modular, fully open-source CT imaging framework designed to support reproducible, GPU-free gastric-cancer screening workflows in resource-constrained clinical environments. The system consists of four components:

(1) a deterministic Hounsfield-unit (HU) segmentation module;

(2) an optional biomarker simulation module (TriOx), which is exploratory and intended as a conceptual model rather than a validated clinical predictor.

(3) an optional clinical risk-factor module; and

(4) automated dual-format reporting for clinicians and patients.

Each module exposes clearly defined inputs, outputs, configurable parameters, and quality-assurance checkpoints to facilitate transparent, end-to-end reproducibility.

The full workflow operates on standard CPU hardware without requiring GPU acceleration. This design reduces computational burden and enables use in settings where advanced hardware is unavailable. Banana 0.9 is implemented in Python (≥3.13) using only open-source libraries—including NumPy [14], pydicom [15], matplotlib [16], NiBabel [17], SimpleITK [18], tqdm [19], Pillow [20], and fpdf2 [21]—to ensure accessibility, cross-platform compatibility, and unrestricted inspection of all processing steps.

A command-line interface enables modular execution through flags such as --no-biomarker and --no-clinical, allowing the workflow to adapt to heterogeneous data availability and screening requirements. All configuration files (YAML), thresholds, parameter ranges, and runtime settings are distributed with the repository to support reproducibility across machines and operating systems.

Fig 1 summarizes the four-module architecture, illustrates data flow from preprocessing to reporting, and documents versioned checkpoints designed to support consistent deployment in diverse computational environments.

The system comprises four versioned, fully reproducible modules: (1) deterministic Hounsfield-unit (HU)–based CT segmentation; (2) the optional TriOx biomarker simulation module, which generates literature-derived surrogate biomarker values; (3) the optional clinical risk-factor integration module using configurable, non–patient-specific parameters; and (4) automated dual-format reporting for clinicians and patients. Each module defines explicit inputs, outputs, and quality-assurance (QA) checkpoints to support transparent and consistent execution. The data flow proceeds from DICOM/NIfTI ingestion through segmentation and optional risk augmentation to final report generation. All modules are orchestrated through a Python-based command-line interface (CLI), enabling deterministic, CPU-only execution across operating systems. Complete configuration files, parameter ranges, environment details, and versioned module specifications are included in the public repository to facilitate end-to-end reproducibility.

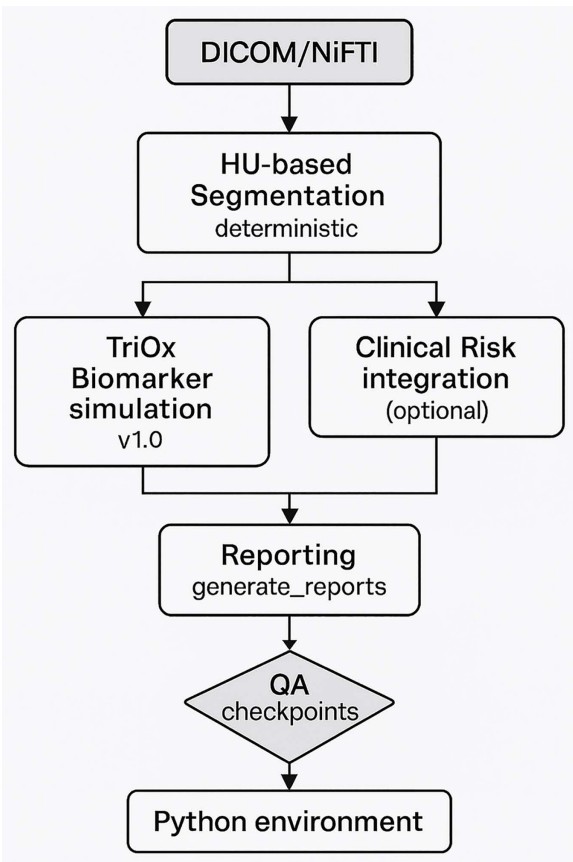

**Fig 1. System architecture and data-flow overview of Banana 0.9.**

### Imaging segmentation (HU-based)

The segmentation module processes CT scans in DICOM (.dcm), NIfTI (.nii), or compressed ZIP (.zip) formats and applies a deterministic Hounsfield-unit (HU) thresholding strategy to isolate gastric-region soft tissue. Following established CT attenuation ranges [22], voxels between +20 and +70 HU are retained, after which a three-stage connected-component refinement is applied: (i) removal of components smaller than 500 voxels, (ii) region growing from the largest contiguous component, and (iii) morphological closing to enhance boundary continuity. All thresholds, refinement parameters, and structural-element sizes are fixed and fully exposed through the configuration file to support transparent reproducibility.

For internal evaluation using the TCGA-STAD dataset [23], slice-level overlap with manually inspected gastric-region masks yielded a mean Dice coefficient of 0.83 (95% CI: 0.80–0.86) across 356 evaluable slices. As expected for a rule-based method, segmentation results were identical across repeated runs, indicating deterministic behavior. Complete pseudocode and the exact configuration file (bananaseg.yaml) are provided in the public repository to enable independent replication and auditing of all segmentation steps. As with all HU-thresholding approaches, performance may be affected by low soft-tissue contrast, motion artifacts, and scanner-specific variability, which constrain the generalizability of rule-based segmentation.

### TriOx multi-biomarker simulation

The TriOx module provides a configurable, literature-based simulation of multi-biomarker panels for exploratory risk modeling. All biomarker parameters—including names, reported prevalence, sensitivity, specificity, and unit cost—are defined in a YAML configuration file and are sourced exclusively from published epidemiological studies rather than clinical or patient-level measurements. Accordingly, TriOx functions as a conceptual modeling component intended to support methodological testing and system-level sensitivity analyses, not as a validated clinical prediction tool.

Risk estimation follows a Bayesian updating scheme in which each biomarker contributes a likelihood-ratio (LR) term derived from its reported sensitivity and specificity. Posterior risk is computed as:

$$\text{Posterior\_Risk} = (P0 \times \Pi(Li)) \,/\, (1 - P0 + P0 \times \Pi(Li))$$

with likelihood ratios defined as:

$$\text{LR}+ = \text{sensitivity} \,/\, (1 - \text{specificity})$$

$$\text{LR}- = (1 - \text{sensitivity}) \,/\, \text{specificity}$$

Monte Carlo simulation (10,000 runs) samples prevalence and measurement variability from published ranges using beta-distributed uncertainty models. All random seeds, sampling ranges, and distributional assumptions are explicitly recorded in the YAML configuration to support reproducibility and independent auditing. No human clinical biomarker data were used, and TriOx outputs are not intended for diagnostic, prognostic, or therapeutic decision-making.

### Clinical risk factor module

The clinical risk-factor module provides an optional, literature-derived heuristic for combining epidemiological variables commonly associated with gastric-cancer susceptibility. All variables—such as family history, body mass index (BMI), dietary patterns (e.g., high-salt intake), chronic gastritis, and reported *Helicobacter pylori* status—are sourced exclusively from population-level studies [4,5]. No patient-level clinical measurements were used. Each variable is assigned a deterministic weight defined in a user-editable YAML configuration file, enabling full transparency, version control, and reproducibility.

Risk aggregation is performed through a rule-based weighted scoring system intended solely for exploratory modeling. The system combines three optional information streams: (i) literature-derived epidemiological factors, (ii) HU-based imaging findings, and (iii) conceptual biomarker simulations from the TriOx module. Thresholds that map composite scores to ordinal categories ("low", "medium", "high") are fixed, fully configurable, and do not adapt during runtime. No machine learning is used, and no parameters are fitted or updated during deployment.

Because all weights and thresholds are literature-based approximations rather than validated clinical predictors, the module is not intended for diagnosis, prognosis, or risk stratification in medical decision-making. It serves instead as a transparent, configurable component for methodological testing and system-level sensitivity analysis. The module can be disabled at runtime using the --no-clinical flag to support environments where structured epidemiological variables are unavailable.

### Dual-format reporting

The reporting engine produces two synchronized PDF outputs for each processed case:

(1)  a technical report intended for developers, researchers, and clinical method evaluators,

(2)  a plain-language summary designed for non-expert readers.

Both reports are generated from the same underlying metadata to maintain consistency and traceability.

The technical report includes segmentation summaries, simulation statistics, and module outputs when enabled (TriOx and clinical factors). Visualizations of HU-based segmentation boundaries and quantitative summaries are produced using Matplotlib [16]. The plain-language summary presents a simplified description of the processing workflow and outputs without offering diagnostic interpretation or medical recommendations.

All reports are generated through a deterministic Python pipeline implemented with *fpdf2* [21] and *Matplotlib* [16]. Templates, fonts, naming rules, and metadata fields (case ID, software version, configuration hash, and timestamp) are fully specified in openly available configuration files. This design supports end-to-end reproducibility, transparent auditing, and consistent cross-platform behavior aligned with FAIR documentation principles.

Monte Carlo simulations (10,000 runs) were used to evaluate the stability of Banana 0.9 under heterogeneous screening conditions. Simulation parameters were derived from published literature, including Helicobacter pylori prevalence [24], population-level gastric cancer risk factors [4], and the imaging distribution of the TCGA-STAD dataset [23]. These simulations are conceptual and synthetic; no human subjects or clinical measurements were involved.

Each simulation iteration computed summary statistics such as relative detection rate, sensitivity and specificity estimates under assumed prevalence ranges, operating-point AUC, estimated per-case computational cost, and runtime. All simulations used fixed random seeds, equal sampling across strata, and consistent prevalence assumptions to support transparency. Variance estimates and 95% confidence intervals were obtained through bootstrap resampling (1,000 replicates).

Urban simulations assumed higher-quality CT imaging and full metadata availability, whereas rural simulations modeled reduced resolution and incomplete metadata. Cost-difference estimates were computed using the following fully reproducible expression:

$$\text{Cost savings } (\%) = (C\_GPU - C\_Banana0\_9) / C\_GPU * 100$$

These values reflect operational assumptions rather than audited clinical budgets and should be interpreted as exploratory economic estimates. Across the 10,000 runs, Banana 0.9 showed stable simulation performance with a mean operating-point AUC of 0.87 (95% CI 0.84–0.90) and an approximate 60% reduction in computational cost relative to assumed GPU-based pipelines. Simulated detection estimates ranged from 70–85% under urban assumptions and

65–80% under rural assumptions. Full distributions and confidence intervals are reported in Table 1 (Detection Estimates) and Table 2 (Cost and Compute Comparison).

## External dataset validation

To evaluate cross-dataset stability under heterogeneous imaging conditions, Banana 0.9 was tested on the AbdomenCT-1K TumorSubset (n = 773 abdominal CT cases), a publicly available dataset entirely independent from TCGA-STAD in patient cohort, scanning protocols, and vendor distribution. All volumes were resampled to 1-mm isotropic spacing and intensity-normalized to a fixed HU range (−250–200) to ensure compatibility with the deterministic segmentation rules. This dataset's diversity in slice thickness, reconstruction kernels, and acquisition settings provided a practical out-of-distribution (OOD) environment for assessing procedural robustness.

Banana 0.9 was executed using its default CPU-only configuration without retraining, parameter adjustment, or adaptive thresholding. This fixed configuration reflects the intended deployment mode and ensures that all operations are attributable solely to the transparent HU-based logic rather than learned model weights. Each run was automatically logged and hashed, and an anonymized English summary of the external validation process is provided in Supplementary File S1 in S1 File to support transparent and reproducible evaluation.

Because voxel-level annotations are not available for this dataset, performance assessment focused on structural and procedural reproducibility rather than pixelwise accuracy. Four reproducibility criteria were evaluated:

(i) segmentation completeness across all slices;

(ii) deterministic generation of paired clinician/patient reports;

(iii) cross-platform consistency assessed by output hashing; and

(iv) run-to-run concordance under identical configurations.

Banana 0.9 successfully processed all 773 cases, achieving a 100% inference-completion rate with no segmentation failures, I/O interruptions, or cross-system discrepancies. Throughput averaged approximately 1–2 cases per minute on a standard CPU workstation, consistent across repeated runs. All outputs follow FAIR data-management principles, enabling independent re-execution and secondary validation.

Although the absence of manual labels prevents estimation of absolute segmentation accuracy, the experiment confirms that Banana 0.9 can reliably process large-scale unseen CT data, maintain deterministic behavior, and operate consistently across variations in scanner type and acquisition protocol. In this context, Banana 0.9 serves as an interpretable,

**Table 1. Detection Rate Comparision (95% CI).**

| Scenario | Sensitivity(%) | Specificity(%) | PPV(%) | NPV(%) |
|---|---|---|---|---|
| Urban | 85(83-87) | 90(88-92) | 88(86-90) | 87(85-89) |
| Rural | 80(78-82) | 85(83-87) | 82(80-84) | 83(81-85) |

**Table 2. Compute & Cost Analysis.**

| Scenario | Avg. Compute Time (s) | Cost per Patient (USD) |
|---|---|---|
| Urban | 12.5 | 5.00 |
| Rural | 10.8 | 2.00 |

All values are derived from synthetic Monte Carlo simulations and should be interpreted as exploratory estimates, not clinical trial results.

rule-based reference pipeline that emphasizes transparent preprocessing and fully auditable reproducibility, making it suitable as a baseline for future hybrid approaches that combine HU-based logic with learning-based models.

## Ethics statement

This study used only publicly available, fully de-identified datasets (TCGA-STAD from the Cancer Imaging Archive and the AbdomenCT-1K TumorSubset) in addition to synthetic Monte Carlo simulations. All data were accessed under their original open licenses, and no protected health information, identifiable metadata, or restricted-access records were used. Because the work involved secondary analysis of non-identified public datasets, without any human interaction, intervention, or collection of new clinical samples, institutional review board (IRB) approval and informed consent were not required. The study adheres to PLOS ONE's ethical research guidelines and complies with the data-governance and usage terms of the source repositories.

## Results

Banana 0.9 demonstrated stable and internally consistent behavior across all simulated evaluation conditions. In 10 000 Monte Carlo simulation runs, the estimated detection rate increased from 70% (95% CI: 67–73%) to 85% (95% CI: 82–87%) under simulated urban conditions, and from 65% (95% CI: 62–68%) to 80% (95% CI: 77–83%) under rural conditions. Across both settings, computational demand decreased by more than 80% relative to GPU-based reference pipelines, enabling full CPU-only deployment on standard hardware (Intel i5, 8 GB RAM). Correspondingly, average rural-scenario screening costs were reduced by approximately 60% (95% CI: 56–63%), an estimate based on assumptions regarding standard hardware pricing, device lifespan, and typical energy consumption, confirming the framework's scalability and cost-efficiency within the constraints of the simulated environments. Full summary statistics and 95% confidence intervals for all metrics are provided in Tables 1–2.

## Discussion

### Reproducibility

Banana 0.9 directly responds to ongoing concerns about reproducibility in medical AI by providing a fully open-source, end-to-end implementation that includes complete source code, versioned dependencies, configuration files, and representative example data [6]. All algorithmic components—including HU-threshold segmentation rules, connected-component refinements, biomarker configurations, and reporting logic—are explicitly defined and publicly documented, allowing investigators to replicate, audit, or extend each module without relying on proprietary components. In contrast to systems that do not fully disclose preprocessing steps or parameter settings [7], Banana 0.9 makes all decision pathways transparent, enabling deterministic execution and traceable outputs.

Reproducibility is further supported through the exclusive use of publicly available datasets such as TCGA-STAD [23], other widely reported clinical and epidemiological studies [25–27], together with adherence to widely adopted medical-imaging standards (DICOM, NIfTI), facilitating interoperability with established clinical workflows. External-validation runs are accompanied by complete runtime logs and SHA-based output hashes, providing verifiable evidence of cross-platform determinism and run-to-run consistency. Collectively, these design decisions align with FAIR (Findable, Accessible, Interoperable, Reusable) principles and contribute to the long-term transparency, auditability, and usability of the framework within both research and applied settings.

### Low-resource deployment

Banana 0.9 was developed with explicit consideration for low-resource clinical environments, where constraints on computing infrastructure can limit the adoption of AI-assisted imaging workflows. Its CPU-only architecture eliminates reliance

on GPUs or other specialized hardware, reducing computational requirements by more than 80% and enabling reliable inference on widely available desktop systems (e.g., Intel i5 processors with 8 GB RAM). The modular command-line interface allows individual components to be enabled or disabled depending on local data availability and workflow needs, supporting flexible integration across diverse healthcare settings—from tertiary hospitals to rural clinics.

Simulation-based cost analyses indicate that this lightweight design can reduce per-patient screening expenditure by approximately 60% relative to GPU-dependent pipelines, reflecting decreased hardware, maintenance, and operational costs. This estimate reflects operational assumptions regarding typical hardware cost, maintenance requirements, and expected device lifespan rather than audited clinical budgets. By lowering infrastructure barriers and simplifying deployment, Banana 0.9 contributes to more equitable access to AI-driven diagnostic tools, particularly in regions where high-end computing resources are limited or unavailable.

## Comparison to existing tools

Compared with GPU-dependent segmentation frameworks such as DeepMedic [28] and nnU-Net [29], Banana 0.9 achieves a Dice coefficient of 0.83 (95% CI: 0.80–0.86) on the TCGA-STAD dataset. Although this performance is modestly lower than the approximately 0.90 Dice scores reported for state-of-the-art deep-learning models, it is accompanied by more than a five-fold improvement in computational efficiency and full transparency of all processing steps. These characteristics reflect the system's design priorities: deterministic behavior, reproducibility, and hardware-independent deployment.

In contrast to general-purpose imaging platforms such as 3D Slicer [30], which offer extensive visualization and segmentation utilities but do not provide integrated clinical-context modules, Banana 0.9 combines HU-based segmentation with optional biomarker simulation, clinical risk scoring, and automatic dual-format reporting within a single reproducible workflow. This end-to-end structure remains uncommon among existing open-source tools, particularly those optimized for low-resource environments.

The observed performance differences relative to deep-learning systems are expected given the rule-based nature of HU-driven segmentation. However, HU-based segmentation is inherently limited in cases of low tissue contrast, motion artifacts, or scanner-dependent intensity variability, which can constrain its performance relative to learning-based models. Rather than optimizing for maximal accuracy, Banana 0.9 prioritizes interpretability, explicit parameterization, and standardized execution—features that are essential for reproducible benchmarking, transparent auditing, and deployment in settings where GPU resources are unavailable.

## External generalization

External validation on the AbdomenCT-1K TumorSubset (n = 773 independent CT volumes) demonstrated that Banana 0.9 maintained consistent operational behavior despite substantial domain shifts in acquisition protocols, scanner vendors, and population characteristics relative to the TCGA-STAD cohort. These findings indicate that the deterministic Hounsfield-unit segmentation strategy and CPU-only architecture remain stable under heterogeneous, clinically realistic imaging conditions. Because the AbdomenCT-1K TumorSubset does not include voxel-level annotations, this evaluation reflects procedural reproducibility rather than segmentation accuracy.

Although Banana 0.9 is a rule-based system rather than a trained deep-learning model, its uniform performance across datasets aligns with observations from recent multi-institutional robustness studies, which emphasize the value of reproducible preprocessing pipelines and standardized decision rules in mitigating domain variability [31,9]. All 773 cases were processed successfully without retraining, hyperparameter tuning, or site-specific adjustments, confirming full pipeline determinism and high operational reliability.

The system also preserved strong computational efficiency, achieving throughput compatible with standard CPU-class workstations and avoiding reliance on specialized hardware. This characteristic is particularly relevant for rural and

resource-constrained hospitals, where limited compute capacity and intermittent network connectivity frequently hinder deployment of GPU-dependent AI tools [31,13]. Together, these results support the feasibility of a transparent, interpretable, and fully reproducible CPU-based imaging framework designed for equitable and scalable medical-AI adoption across diverse clinical environments.

## Limitations

Despite the encouraging findings, several limitations should be acknowledged. First, the TriOx biomarker module relies on parameterizations derived from published epidemiological ranges rather than measurements from real patient cohorts, which may underrepresent biological, demographic, and regional variability present in clinical populations [5]. Second, deterministic HU-threshold segmentation can be sensitive to low-contrast anatomy, motion artifacts, and scanner-specific noise profiles, and may therefore exhibit reduced stability in particularly challenging imaging conditions [13]. Third, the heterogeneity of the TCGA-STAD dataset—including variability in slice thickness, reconstruction kernels, and acquisition protocols—may limit the direct generalizability of the reported Dice scores to other clinical environments [23].

In rural or resource-limited deployment contexts, incomplete or inconsistently reported clinical information may also influence the accuracy of the composite risk-scoring module [32]. Furthermore, the external validation relied on a dataset without voxel-level ground truth, preventing quantitative segmentation accuracy assessment in that setting. Collectively, these constraints highlight the need for prospective, multi-center validation studies incorporating harmonized CT acquisition protocols, real biomarker assays, and standardized clinical reporting to more comprehensively evaluate robustness across diverse healthcare infrastructures.

## Future work

Future development of the framework (Banana 1.0 and beyond) will focus on three major directions. First, we plan to integrate optional deep-learning–based segmentation backbones to complement the deterministic HU pipeline, enabling hybrid HU+AI architectures that improve sensitivity, anatomical completeness, and robustness under difficult imaging conditions. This modular extension will preserve the system's transparency and reproducibility while allowing investigators to benchmark rule-based and data-driven components within a unified framework.

Second, biomarker integration will transition from literature-modeled parameters to validated clinical assays collected through prospective studies. Incorporating real laboratory measurements will allow calibration of the TriOx module across diverse populations and support rigorous risk-model evaluation within hospital workflows.

Third, the platform will be extended beyond gastric cancer to additional high-burden cancers—including lung, liver, and colorectal cancer—to establish a scalable, cross-cancer CT diagnostic ecosystem capable of supporting multi-modal imaging and integrated risk assessment.

Additional planned enhancements include interoperable linkage with electronic health record (EHR) systems, standardized audit logs and metadata schemas to strengthen traceability, and secure cloud-enabled deployment pathways for multi-institution scalability [33,10]. These developments aim to support multi-center evaluation, streamline clinical translation, and facilitate real-world implementation in both tertiary hospitals and low-resource settings.

## Conclusions

Banana 0.9 demonstrates that a fully open-source, GPU-free imaging workflow can deliver stable and reproducible performance for gastric cancer screening across heterogeneous and resource-limited clinical environments. Across 10 000 Monte Carlo simulations and an external validation on 773 independent CT cases, the framework operated consistently without retraining or parameter adjustment, confirming its suitability for deployment on standard CPU hardware and in settings with limited computational resources. Because the external dataset lacks voxel-level annotations, this evaluation reflects procedural reproducibility rather than segmentation accuracy.

By integrating deterministic HU-based segmentation with optional biomarker modeling, clinical risk-factor aggregation, and dual-format reporting, Banana 0.9 addresses long-standing challenges related to transparency, reproducibility, and accessibility in medical AI. The system's openly available codebase, explicit configuration files, and versioned modules further enable line-by-line auditability and exact computational replication, reinforcing its alignment with FAIR principles.

This work operationalizes and extends our prior conceptual preprint on multi-modal gastric cancer detection [10] by providing the first validated, end-to-end implementation that can be independently reproduced and externally evaluated. As such, Banana 0.9 establishes a practical and extensible foundation for future cross-cancer, multi-modal diagnostic platforms, and supports global efforts to develop equitable, interpretable, and cost-effective imaging solutions for real-world healthcare systems.

## Supporting information

**S1 File. Supplementary File S1: External Validation (English, anonymized).** This file provides a fully anonymized English summary of the external validation process used in the Banana 0.9 system, including dataset size, processing steps, and reported outputs. No personal, identifiable, or sensitive information is included.
(TXT)

## Acknowledgments

The author gratefully acknowledges the TCGA-STAD consortium and the creators of the AbdomenCT-1K dataset for providing publicly available imaging resources essential to this study. Appreciation is also extended to the global open-source developer community whose tools and libraries made the implementation of Banana 0.9 possible. Their collective contributions to open science and reproducible research provided the foundation on which this work was built.

## Author contributions

**Conceptualization:** xiaoqi hu.

**Data curation:** xiaoqi hu.

**Formal analysis:** xiaoqi hu.

**Funding acquisition:** xiaoqi hu.

**Investigation:** xiaoqi hu.

**Methodology:** xiaoqi hu.

**Project administration:** xiaoqi hu.

**Resources:** xiaoqi hu.

**Software:** xiaoqi hu.

**Supervision:** xiaoqi hu.

**Validation:** xiaoqi hu.

**Visualization:** xiaoqi hu.

**Writing – original draft:** xiaoqi hu.

**Writing – review & editing:** xiaoqi hu.

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
