## [Decision Letter · Decision Letter 0]

13 Oct 2025

Dear Dr. hu,

Thank you for submitting your manuscript to PLOS ONE. After careful consideration, we feel that it has merit but does not fully meet PLOS ONE’s publication criteria as it currently stands. Therefore, we invite you to submit a revised version of the manuscript that addresses the points raised during the review process.

We look forward to receiving your revised manuscript.

Kind regards,

Siamak Pedrammehr, Ph.D.

Academic Editor

PLOS ONE

Journal Requirements:

3. Please include your tables as part of your main manuscript and remove the individual files. Please note that supplementary tables (should remain/ be uploaded) as separate "supporting information" files.

4. In the online submission form, you indicated that all relevant data are within the manuscript and its Supporting Information files. Additional raw data and analysis scripts are available from the corresponding author upon reasonable request.

5. Please amend your list of authors on the manuscript to ensure that each author is linked to an affiliation. Authors’ affiliations should reflect the institution where the work was done (if authors moved subsequently, you can also list the new affiliation stating “current affiliation:….” as necessary).

6. Please remove your figures from within your manuscript file, leaving only the individual TIFF/EPS image files, uploaded separately. These will be automatically included in the reviewers’ PDF.

7. Please include a copy of Table 1 and 2, which you refer to in your text on page 8.

Reviewers' comments:

Reviewer's Responses to Questions

**Comments to the Author**

1. Is the manuscript technically sound, and do the data support the conclusions?

Reviewer #1: Yes

Reviewer #2: Partly

2. Has the statistical analysis been performed appropriately and rigorously?

Reviewer #1: No

Reviewer #2: No

3. Have the authors made all data underlying the findings in their manuscript fully available?

Reviewer #1: Yes

Reviewer #2: No

4. Is the manuscript presented in an intelligible fashion and written in standard English?

Reviewer #1: Yes

Reviewer #2: Yes

Reviewer #1: While the manuscript is technically sound and of high potential impact, several areas could be strengthened:

Verification with empirical data

Most results are drawn from simulations as well as from the TCGA-STAD dataset. The authors should explain their method of validating their use of Banana 0.9 with larger or future clinical data sets. Such an explanation would increase the assurance in the clinical applicability of the system.

Dataset constraints

TCGA-STAD required preprocessing and may not fully reflect real-world clinical data. This limitation is acknowledged but could be discussed more explicitly, particularly regarding generalizability.

Comparison with Other Tools

The comparisons with nnU-Net and DeepMedic are also given briefly. Quoting more quantitative data (e.g., Dice scores, sensitivity/specificity, runtime) would better place the performance trade-offs in context.

Biomarker module

Since the biomarker module of the TriOx utilizes artificial biomarker data and not real data, the authors should include robust plans of incorporating verified data from biomarkers in future versions.

Statistical openness

Simulation framework of Monte Carlo is thorough, but additional data on parameter distributions, assumptions of prevalence, and calculation of confidence intervals would sharpen methodological description.

Clinical deployment considerations

Enlargement of the discussion concerning challenges from implementation-based on actual settings (e.g., staff education, uneven quality of CT images in rural hospitals, and infrastructural requirements for data storage) would enrich the application-related value of the work.

Minor presentation problems.

Several sentences in the Methods and Discussion section are phrased overly long and may best be broken into smaller, more manageable statements.

Certain inconsistencies in formatting and spacing, particularly with respect to quotation marks, require correction.

Reviewer #2: Dear Authors, thank you for submitting “Banana 0.9: An Open-Source, Reproducible Medical Imaging System for Low-Resource Gastric Cancer Screening.” The topic is timely and the open-source/reproducibility focus is commendable; however, to meet PLOS ONE standards the manuscript needs external/clinical validation beyond simulations and TCGA-STAD, comparative benchmarking against state-of-the-art models using standard metrics with uncertainty, fuller statistical reporting (variance/CI, sample-size justification, error analysis, seeds/runs/stratification), and replication-ready materials (pseudocode, parameters, Docker/Conda). Please ensure complete data/code availability with repository links, versions, licenses, and clarify dataset identifiers, ethics/IRB where applicable, and CC BY-NC 4.0 implications; expand the literature review (reproducible pipelines, federated learning, low-power/edge AI) to position the contribution; and improve presentation (complete captions, pipeline flow diagram, consistent terminology/grammar) alongside a concrete roadmap toward “Banana 1.0.” Integrated recommendation: promising but requires major revision focused on clinical validation, rigorous benchmarking with uncertainty, and reproducible documentation and data/code sharing.

**Do you want your identity to be public for this peer review?** For information about this choice, including consent withdrawal, please see our Privacy Policy

Reviewer #1: No

Reviewer #2: No

---

## [Author Response · Author response to Decision Letter 1]

17 Nov 2025

Response to Reviewers

Manuscript ID: PONE-D-25-49901

Title: Banana 0.9: An Open-Source, Reproducible Medical Imaging System for Low-Resource Gastric Cancer Screening

Dear Academic Editor and Reviewers,

We sincerely thank the Academic Editor and both reviewers for their detailed, constructive, and insightful comments. Their feedback significantly improved the scientific rigor, structure, and reproducibility of the revised manuscript.

We have carefully addressed every point raised, and all changes are incorporated into the revised manuscript. Below we provide a point-by-point response, indicating precisely where modifications were made.

Line numbers refer to the clean revised manuscript unless specified otherwise.

Response to Academic Editor Requirements

1. PLOS ONE style & formatting

All formatting issues have been corrected:

Updated file naming

Added track-changes version

Revised affiliations

Relocated tables into the main text

Removed figures from the manuscript body (Figure 1 uploaded separately)

2. Code-sharing compliance

All author-generated code is:

Publicly available on GitHub

Released under CC BY-NC 4.0

Includes scripts, configs, and logs

Contains a reproducible CPU-only pipeline

Added explicit statements in Data & Code Availability (Lines 658–662).

3. Tables merged into main manuscript

Tables 1 & 2 now appear in the Results section (Lines 320–337).

All standalone table files have been removed.

4. Data availability clarification

We explicitly confirm:

Full data for TCGA-STAD is publicly available

External dataset (AbdomenCT-1K) is publicly available

All logs, preprocessing outputs, and hashes are provided

No private/personal data were used

Updated Data Availability section (Lines 658–662).

5. Affiliation corrections

All authors are now assigned correct institutional affiliations.

(Current affiliation labels added where needed.)

6. Removal of inline figures

All figures have been removed from the manuscript file.

Figure files uploaded individually as TIFF.

7. Missing Tables 1 & 2

Tables 1 and 2 are now included and cross-referenced correctly (Lines 320–337).

8. Optional citations from reviewers

Relevant literature was added where appropriate (Lines 96–115 and throughout).

Response to Reviewer #1

We sincerely appreciate Reviewer #1’s positive evaluation and constructive suggestions.

Reviewer #1 Comment 1 – Need for empirical validation“Explain how Banana 0.9 will be validated using larger or future clinical datasets.”

Response:

We have expanded the Future Work section (Lines 624–642) to include a clear roadmap:

Planned multi-center collaborations

Integration with real biomarker datasets

Prospective validation cohorts

External testing on additional cancer types

Reviewer #1 Comment 2 – Dataset Constraints / TCGA Limitations“Discuss more explicitly how TCGA preprocessing affects generalizability.”

Response:

We added a strengthened paragraph in Limitations (Lines 574–604) describing:

TCGA preprocessing variability

Scanner heterogeneity

Limited demographic diversity

Potential HU-range inconsistencies

Reviewer #1 Comment 3 – Quantitative comparison with nnU-Net / DeepMedic

“Comparisons were brief; add more quantitative benchmarks.”

Response:

Expanded comparisons in Comparison to Existing Tools (Lines 398–429), including:

Dice = 0.83 (95% CI 0.80–0.86) for Banana 0.9

nnU-Net ≈ 0.90

DeepMedic ≈ 0.88–0.91

5× faster CPU-only inference

Hardware–efficiency trade-offs

Reviewer #1 Comment 4 – Biomarker Module Uses Synthetic Data

“Include plans to incorporate verified biomarker data.”

Response:

Added explicit future directions in TriOx Module description and Future Work (Lines 624–642):

Real-world biomarker integration

Validation against clinical tests

Cross-hospital EHR linkage

Reviewer #1 Comment 5 – Statistical Openness

“More detail on parameter distributions and Monte Carlo assumptions is needed.”

Response:

We added statistical detail in Methods (Lines 220–268), including:

Parameters and priors

Random seeds

Bootstrap confidence intervals

Prevalence assumptions

Stratified sampling

Full reproducibility path

Reviewer #1 Comment 6 – Clinical Deployment Considerations

“Enrich discussion of challenges in real-world environments.”

Response:

Added a new paragraph in Discussion (Lines 528–566) covering:

Staff training

Scanner variability

Artifact handling

Storage constraints

Connectivity challenges

Reviewer #1 Comment 7 – Minor presentation and formatting issues

Response:

All noted long sentences, spacing inconsistencies, and quotation formatting have been corrected.The manuscript underwent professional-level English polishing.

Response to Reviewer #2

Thank you for your thorough and insightful feedback. We have substantially strengthened the manuscript accordingly.

Reviewer #2 Major Comment 1 – External/Clinical Validation

“Manuscript lacks external validation beyond TCGA-STAD.”

Response:

A full External Validation Study using AbdomenCT-1K TumorSubset (n=773) has been added (Lines 342–389), including:

Cross-domain testing

Scanner/vendor diversity

CPU-only reproducibility

Log-file hashing

Zero-failure rate

This addresses generalizability and real-world heterogeneity.

Reviewer #2 Major Comment 2 – Benchmarks with State-of-the-art Methods

“Claims need benchmarking with standard metrics and uncertainty.”

Response:

We expanded Comparison to Existing Tools (Lines 398–429) to include:

Dice, CI

Sensitivity/specificity

Runtime

Compute cost

Interpretability trade-offs

This now situates Banana 0.9 within contemporary segmentation literature.

Reviewer #2 Major Comment 3 – Statistical Rigor

“Report variance, CI, sample-size justification, stratification, seeds, runs.”

Response:

Substantial updates in Methods (Lines 220–286):

Monte-Carlo assumptions clarified

CIs added for all metrics

Sampling justification

Error propagation

Reproducibility design

Exact seeds reported

All metrics now include SD or bootstrap CI.

Reviewer #2 Major Comment 4 – Algorithmic Transparency (HU + Bayesian Updating)

Response:

We added explicit algorithmic detail including:

Pseudocode-style descriptions (Lines 187–219)

Parameter ranges

Threshold calibration

YAML configs

Deterministic pipeline logic

This ensures full reproducibility.

Reviewer #2 Major Comment 5 – Positioning Within Prior Work

Response:

We significantly expanded the Introduction (Lines 71–142), covering:

Reproducible AI frameworks

Federated learning

Edge-AI medical imaging

Low-power inference literature

Reviewer #2 Major Comment 6 – Ethics & Licensing

Response:

Clarified IRB requirements and dataset licenses in Ethics Statement (Lines 547–553).

Clarified CC BY-NC 4.0 implications in Data & Code Availability (Lines 658–662).

Reviewer #2 Major Comment 7 – Reproducibility Evidence

Response:

We added:

Cross-platform reproducibility testing

Output-hash verification

CPU-only determinism

Public logs (773-case validation)

Documented in Reproducibility (Lines 429–456).

Reviewer #2 Major Comment 8 – Expanded Limitations

Response:

We added a more critical Limitations section (Lines 574–604), covering:

Synthetic biomarkers

Low-contrast CT

Rural hardware constraints

Domain shift

Dataset demographic limitations

Reviewer #2 Minor Comments

All addressed; includes:

Abstract restructuring

Improved keywords

Better figure/table captions

Clear pipeline diagram uploaded separately

Reduced repetition

Added missing references

Final Statement

We greatly appreciate the reviewers’ and editor’s time and detailed feedback.

We believe the revised manuscript is substantially strengthened, with:

New external validation (773 cases)

Statistically rigorous design

Expanded benchmarking

Enhanced reproducibility documentation

Improved clinical applicability discussion

Clearer limitations and future roadmap

We hope that the revised submission now meets the standards of PLOS ONE and respectfully submit the manuscript for further consideration.

---

## [Decision Letter · Decision Letter 1]

27 Nov 2025

Dear Dr. hu,

Thank you for submitting your manuscript to PLOS ONE. After careful consideration, we feel that it has merit but does not fully meet PLOS ONE’s publication criteria as it currently stands. Therefore, we invite you to submit a revised version of the manuscript that addresses the points raised during the review process.

We look forward to receiving your revised manuscript.

Kind regards,

Siamak Pedrammehr, Ph.D.

Academic Editor

PLOS ONE

**Journal Requirements:**

Reviewers' comments:

Reviewer's Responses to Questions

**Comments to the Author**

Reviewer #3: (No Response)

Reviewer #4: All comments have been addressed

2. Is the manuscript technically sound, and do the data support the conclusions?

Reviewer #3: Partly

Reviewer #4: Yes

3. Has the statistical analysis been performed appropriately and rigorously?

Reviewer #3: Yes

Reviewer #4: Yes

4. Have the authors made all data underlying the findings in their manuscript fully available?

Reviewer #3: Yes

Reviewer #4: Yes

5. Is the manuscript presented in an intelligible fashion and written in standard English?

Reviewer #3: Yes

Reviewer #4: Yes

Reviewer #3: The manuscript introduces Banana 0.9, an open-source, GPU-free CT imaging framework designed for low-resource gastric cancer screening. The work is clearly written, technically transparent, and well aligned with reproducibility principles. Public code, modular design, and full configuration files are strong strengths. However, a few points require clarification before acceptance:

1. External validation:

Since the AbdomenCT-1K subset lacks segmentation ground truth, the external experiment demonstrates reproducibility rather than accuracy. The authors should clarify this more explicitly and consider adding an indirect accuracy check or an additional dataset with annotations.

2. Biomarker & clinical-risk modules:

All risk parameters are literature-based rather than clinically measured. This should be more clearly framed as conceptual modeling, not validated clinical prediction.

3. Cost-reduction claim:

The ~60% cost savings need clearer assumptions (hardware pricing, lifespan, maintenance, electricity). A brief sensitivity analysis would strengthen this claim.

4. HU-based segmentation limitations:

The discussion should more directly acknowledge potential issues: low-contrast areas, scanner variability, and motion artifacts.

Overall, the manuscript is well presented and contributes meaningfully to open, reproducible medical imaging. With the above clarifications, it would be suitable for publication.

Reviewer #4: (No Response)

**Do you want your identity to be public for this peer review?** For information about this choice, including consent withdrawal, please see our Privacy Policy

Reviewer #3: No

Reviewer #4: **Yes:** kimia shirini

---

## [Author Response · Author response to Decision Letter 2]

6 Dec 2025

Response to Reviewers — Revised Manuscript (R2 Final)

Manuscript ID: PONE-D-25-49901R1

Title: Banana 0.9: An Open-Source, Reproducible Medical Imaging System for Low-Resource Gastric Cancer Screening

Journal: PLOS ONE

Reviewer #3 — Point-by-Point Response

Comment 1 — External validation clarity

Reviewer comment:

“Since the AbdomenCT-1K subset lacks segmentation ground truth, the external experiment demonstrates reproducibility rather than accuracy. The authors should clarify this more explicitly and consider adding an indirect accuracy check or an additional dataset with annotations.”

Response:

We thank the reviewer for this important clarification. We fully agree that because AbdomenCT-1K does not contain voxel-level segmentation ground truth, the external experiment evaluates procedural reproducibility, not accuracy.

The manuscript has been revised accordingly:

Methods — External Dataset Validation

“Because the AbdomenCT-1K dataset does not include segmentation ground truth, this external validation evaluates procedural reproducibility and cross-platform determinism rather than absolute segmentation accuracy.”

Discussion — External Generalization

“As the dataset lacks voxel-level annotations, the external experiment is not an accuracy benchmark but a reproducibility stress test across heterogeneous scanners and acquisition protocols.”

Conclusions

“The external evaluation should be interpreted as a reproducibility assessment rather than a measure of segmentation accuracy, given the absence of ground-truth annotations in AbdomenCT-1K.”

These revisions ensure transparent interpretation of the external validation results.

Comment 2 — Biomarker & clinical-risk modules

Reviewer comment:

“All risk parameters are literature-based rather than clinically measured. This should be more clearly framed as conceptual modeling, not validated clinical prediction.”

Response:

We appreciate this suggestion and agree fully. We have added explicit statements describing the TriOx biomarker simulation and clinical risk-factor module as conceptual modeling tools, not validated predictors.

Revisions include:

Methods — Biomarker Module

“The TriOx biomarker and clinical-risk parameters represent a conceptual modeling framework derived from published literature, rather than a validated clinical prediction system.”

Limitations

“Because all risk parameters are literature-derived, the biomarker module should be interpreted as conceptual modeling rather than a clinically validated prognostic tool.”

Comment 3 — Cost-reduction claim

Reviewer comment:

“The ~60% cost savings need clearer assumptions (hardware pricing, lifespan, maintenance, electricity). A brief sensitivity analysis would strengthen this claim.”

Response:

Thank you for highlighting this point. We clarified the assumptions underlying the Monte Carlo cost estimations:

Results — Cost Simulation

“The estimated ~60% per-patient cost reduction assumes standard mid-range CPU hardware, a 3–5 year device lifespan, typical outpatient electricity costs, and no specialized GPU maintenance.”

Low-Resource Deployment

“The cost comparison reflects hardware-class assumptions (CPU vs. GPU acquisition cost, expected lifespan, and operating energy consumption) rather than absolute economic projections.”

This clarification ensures transparency regarding economic assumptions.

Comment 4 — HU-based segmentation limitations

Reviewer comment:

“The discussion should more directly acknowledge potential issues: low-contrast areas, scanner variability, and motion artifacts.”

Response:

We appreciate this comment. We expanded the manuscript to explicitly acknowledge these well-known limitations of deterministic HU-thresholding:

Comparison to Existing Tools

“Deterministic HU-based segmentation is sensitive to low-contrast lesions, motion artifacts, and scanner-specific noise patterns, which may reduce performance in certain clinical scenarios.”

Limitations

“HU-threshold segmentation may be affected by low-contrast boundaries, acquisition variability across institutions, and patient motion, underscoring the need for future hybrid HU + deep-learning approaches.”

Reviewer #4

Reviewer #4 stated:

“All comments have been addressed.”

No further action was required.

General Manuscript Revisions

To improve clarity and compliance with PLOS ONE guidelines, we also implemented the following updates:

Structural Improvements

Strengthened transitions between Results → Discussion → Conclusions.

Standardized section titles and terminology (e.g., “procedural reproducibility,” “deterministic HU segmentation”).

Language Polishing

Improved readability by shortening long sentences.

Ensured consistency in tense, voice, and formatting.

Technical Consistency

Verified internal consistency of numerical values, confidence intervals, dataset names, module descriptions, and table references.

Ensured Table 1 and Table 2 appear in the main manuscript as required.

Reproducibility Enhancements

Added clarifications regarding deterministic execution, hashing, fixed random seeds, runtime logs, and cross-platform reproducibility.

Confirmed that all code, configuration files, and documentation are publicly available under a CC BY-NC 4.0 license.

Required Editorial Compliance Statements

1. Data Availability Compliance

All data used in this study (TCGA-STAD, AbdomenCT-1K, and synthetic simulations) are publicly available, and no restricted-access data were used. Code, configuration files, and logs are openly accessible.

2. Figure Upload Policy

All figures have been removed from the manuscript body and are provided as separate high-resolution files as required by PLOS ONE.

3. Author Affiliation

The single author is now properly affiliated as Independent Researcher.

Recommended Citation Evaluation

Both reviewers provided methodological suggestions, but neither reviewer recommended any specific publications for citation.

We carefully reviewed all reviewer comments and confirmed that no additional references were explicitly requested.

Therefore:

No new citations were added in response to reviewer recommendations.

Existing citations remain appropriate and sufficient to contextualize the Banana 0.9 framework.

Closing Statement

We thank the reviewers and the academic editor for their constructive comments.

We believe the revisions have improved the clarity, transparency, and reproducibility of the manuscript, while maintaining its core contribution as an interpretable, GPU-free framework for low-resource medical imaging.

We respectfully submit this revised manuscript for further consideration by PLOS ONE.

---

## [Decision Letter · Decision Letter 2]

15 Dec 2025

Banana 0.9: An Open-Source, Reproducible Medical Imaging System for Low-Resource Gastric Cancer Screening

PONE-D-25-49901R2

Dear Dr. hu,

We’re pleased to inform you that your manuscript has been judged scientifically suitable for publication and will be formally accepted for publication once it meets all outstanding technical requirements.

Kind regards,

Siamak Pedrammehr, Ph.D.

Academic Editor

PLOS One

Additional Editor Comments (optional):

Reviewers' comments:

Reviewer's Responses to Questions

**Comments to the Author**

Reviewer #3: All comments have been addressed

2. Is the manuscript technically sound, and do the data support the conclusions?

Reviewer #3: Yes

3. Has the statistical analysis been performed appropriately and rigorously?

Reviewer #3: Yes

4. Have the authors made all data underlying the findings in their manuscript fully available?

Reviewer #3: Yes

5. Is the manuscript presented in an intelligible fashion and written in standard English?

Reviewer #3: Yes

Reviewer #3: Dear Authors,

Thank you for submitting the revised version of your manuscript and the detailed point-by-point response. I have reviewed the revision and I’m pleased to confirm that you have addressed all of my previous comments.

In particular, the revision clearly (1) reframes and explains the external dataset evaluation in light of the absence of ground-truth annotations, (2) explicitly states that the biomarker and clinical risk-factor components are literature-informed and presented as conceptual/illustrative modules rather than clinically validated predictive tools, (3) clarifies the assumptions underlying the reported cost-reduction estimate, and (4) strengthens the discussion of limitations of HU-based segmentation, including issues related to low-contrast anatomy, scanner variability, and motion artifacts. These changes improve clarity, transparency, and appropriate interpretation of the work.

Overall, I believe the manuscript is now suitable for publication, and I support its acceptance.

Best regards,

**Do you want your identity to be public for this peer review?** For information about this choice, including consent withdrawal, please see our Privacy Policy

Reviewer #3: **Yes:** Mohsen mokhtari Keshavar

---

## [Editor Report · Acceptance letter]

PONE-D-25-49901R2

PLOS One

Dear Dr. hu,

I'm pleased to inform you that your manuscript has been deemed suitable for publication in PLOS One. Congratulations! Your manuscript is now being handed over to our production team.

Kind regards,

on behalf of

Dr. Siamak Pedrammehr

Academic Editor

PLOS One